# Socio-Economic Inequalities in Child Stunting Reduction in Sub-Saharan Africa

**DOI:** 10.3390/nu12010253

**Published:** 2020-01-18

**Authors:** Kaleab Baye, Arnaud Laillou, Stanley Chitweke

**Affiliations:** 1Center for Food Science and Nutrition, Addis Ababa University, PO Box: 1176, Addis Ababa, Ethiopia; 2United Nations Children’s Fund (UNICEF), Addis Ababa, Ethiopia; alaillou@unicef.org (A.L.); schitekwe@unicef.org (S.C.)

**Keywords:** stunting, inequities, sub-Saharan Africa, continuum of care

## Abstract

Stunting in children less than five years of age is widespread in Sub-Saharan Africa. We aimed to: (i) evaluate how the prevalence of stunting has changed by socio-economic status and rural/urban residence, and (ii) assess inequalities in children’s diet quality and access to maternal and child health care. We used data from nationally representative demographic and health- and multiple indicator cluster-surveys (DHS and MICS) to disaggregate the stunting prevalence by wealth quintile and rural/urban residence. The composite coverage index (CCI) reflecting weighed coverage of eight preventive and curative Reproductive, Maternal, Neonatal, and Child Health (RMNCH) interventions was used as a proxy for access to health care, and Minimum Dietary Diversity Score (MDDS) was used as a proxy for child diet quality. Stunting significantly decreased over the past decade, and reductions were faster for the most disadvantaged groups (rural and poorest wealth quintile), but in only 50% of the countries studied. Progress in reducing stunting has not been accompanied by improved equity as inequalities in MDDS (*p* < 0.01) and CCI (*p* < 0.001) persist by wealth quintile and rural-urban residence. Aligning food- and health-systems’ interventions is needed to accelerate stunting reduction more equitably.

## 1. Introduction

Stunting in children less than five years of age is widespread in low and middle income countries (LMIC), with significant proportions of stunted children found in South East Asia and Sub-Saharan Africa [1]. In 2018, >30% of children younger than five years of age were stunted in South Asia and Sub-Saharan Africa. This is significantly higher than in WHO regions like Latin America and the Caribbean (9%) [2]. The causes of stunting can be complex and includes poor maternal health and nutrition, suboptimal infant and young child feeding practices, as well as diseases (e.g., infections); hence, addressing these causes has been given a priority [3]. Stunting has been associated with the compromised growth and development of several organs, including the brain; hence, is linked to poor cognitive and physical performance, which in turn undermines the education, productivity, and future earnings of those affected [4,5,6]. The economic cost of stunting has been estimated to be as high as ~10% of the Gross Domestic Product (GDP) of African countries [7].

Recognizing the largely irreversible consequences of stunting, many African countries have joined the first 1000 days movement that aims to prevent malnutrition from conception to the second birthday of the child [8]. The African Regional Nutrition Strategy (ARNS) adopted by member states of the African Union, and in line with the World Health Assembly, has targets to reduce the number of stunted children younger than five years of age by 40%, by 2025 [9]. Consequently, multiple nutrition-specific/-sensitive interventions have been implemented to varying extents.

Nevertheless, many African countries rely on external support to fund nutrition interventions, and limitations in institutional capacities to effectively implement interventions at scale can also compromise the coverage of key interventions [10]. The extent to which nutrition interventions are reaching those who need them the most remains largely unknown. This is unfortunate, as inequalities in the coverage of essential interventions and thus in stunting reduction, can sustain or even exacerbate long-term inequalities. Therefore, the aim of this study was to evaluate the access of essential health interventions, diet quality, and stunting by wealth quintile and rural/urban residence.

## 2. Methods

### 2.1. Data Sources

The most recent available data were obtained from nationally representative cross-sectional Demographic and Health Surveys (DHS) from Sub-Saharan African countries. The DHS gather data on indicators that can help assess access to health care, child nutrition, and infant and young child feeding practices. For example dietary diversity, meal frequency, and the proportion of children meeting the minimum adequate diet are captured using standardized questionnaires. The DHS uses a multistage stratified sampling design, with households drawn randomly at the last stage.

### 2.2. Data Analyses

#### 2.2.1. Stunting

Stunting was defined as height/length-for-age z-scores <−2 SD relative to the WHO child growth standards [11]. The prevalence of stunting was estimated for children younger than five years of age. The time trends in stunting by urban-rural residence and wealth quintile was presented for countries with at least two surveys spaced between 1998–2008 and 2009–2018. The annual absolute excess change was presented by deducting the percentage point changes in the urban to the rural, and the prevalence in the wealthiest to the poorest. Negative values indicated faster changes in the most disadvantaged group (poorest wealth quintile/rural).

#### 2.2.2. Composite Coverage Index

The Composite Coverage Index (CCI) is a weighted score reflecting the coverage of the following eight preventive and curative Reproductive, Maternal, Neonatal and Child interventions (RMNCH) along the continuum of care—(1) demand for family planning satisfied (modern methods); (2) antenatal care coverage (at least four visits); (3) births attended to by skilled health personnel; (4) BCG immunization coverage among one-year-olds; (5) measles immunization coverage among one-year-olds; (6) DTP3 immunization coverage among one year-olds; (7) children aged less than five years with diarrhea receiving oral rehydration therapy and continued feeding; and (8) children aged less than five years with pneumonia symptoms taken to a health facility [12,13]. The interventions, although not directly linked to nutritional outcomes, are good proxies of access to health care, which is the entry point for most nutrition-specific interventions. For example, vaccination coverage can be a proxy for Vitamin A supplementation, 4+ antenatal care for nutrition counseling, and oral rehydration therapy, and continued feeding during diarrhea can be related to counseling on child feeding during and after sickness. The CCI has been successfully used to track the changes in universal health coverage, but also to monitor the within country socio-economic inequalities [13,14]. Data used to calculate the CCI are derived from the re-analysis of the Demographic and Health Surveys (DHS), Multiple Indicator Cluster Surveys (MICS), and Reproductive Health Surveys (RHS) data, which are publicly available.

#### 2.2.3. Infant and Young Child Feeding Practices

The proportion of infants and young children that are meeting the minimum meal frequency (MMF), minimum dietary diversity (MDD), and minimum acceptable diet (MAD) were calculated using the revised UNICEF/WHO indicators [15]. As part of the DHS survey design, these indicators are collected from the youngest child under two years of age born to mothers aged 15–49 years and from children living with the mother at the time of the survey. The revised UNICEF/WHO indicator counts breastfeeding as one group, thus allowing better comparability between breastfed and non-breastfed children.

#### 2.2.4. Socio-Economic Stratification

Wealth quintiles and place of residence were used as stratification variables in our analyses. DHS uses a wealth index derived using principal component analyses applied to a list of household assets/characteristics, which are country-specific. The first quintile (Q1) represents the 20% poorest families, and the last quintile (Q5) represents the 20% wealthiest families. Quintiles correspond to the relative position of households within each national sample. Urban and rural residence was classified according to boundaries provided by local authorities.

### 2.3. Ethics

All the analyses were based on publicly available data from national DHS surveys. Ethical clearance was the responsibility of the institutions that administered the surveys. The data was obtained after registering in the DHS website, and the datasets did not contain any personal identifiers.

### 2.4. Statistical Analyses

Analyses were done using SPSS version 20. Descriptive statistics were presented as a mean or median for continuous variables, and as a percentage for counts. Inequalities in stunting prevalence and complementary diet quality measures were presented by wealth quintile and rural/urban residence using equiplots (http://www.equidade.org/equiplot.php) generated using STATA version 12. Each horizontal line shows the results by quintile or rural/urban for a given country. Normal distributions of the variables were checked using Kolmogorov-Smirnov test. Independent-*t*-test was used to compare CCI and DDS between rural-urban and poorest-richest wealth quintiles. *p*-values ≤ 0.05 were considered statistically significant.

## 3. Results

### 3.1. Trends in Stunting Prevalence

Stunting prevalence was significantly reduced in most regions of the world, where the highest burden of stunting is reported (Figure 1). Africa has witnessed an average of 18-percentage point reduction in stunting, since 2000. However, Asia, and Latin America and Caribbean regions noticed a much faster reduction in stunting in the 2000–2016 period.

### 3.2. Rate of Stunting Reduction by Rural-Urban Residence and Wealth Quintile

Figure 2 presents the comparisons in stunting reduction by rural-urban among the 30 countries that had at least two surveys in the period of 1998–2018. Our findings show that a relatively faster change in urban than in rural settings in 10/30 countries.

The same rate of stunting reduction was observed by rural/urban residence in 5/30 of the countries. In 15/30 of the countries, a relatively faster pace of change was observed in rural than in urban settings. Figure 3 presents the pace of change in stunting between the poorest and the richest quintiles. An equal number of countries (15/30) had stunting reductions faster in the wealthiest than in the poorest quintile and vice-versa.

### 3.3. Most Recent Estimates of Stunting Prevalence by Rural/Urban Residence and Wealth Quintile

Figure 4 presents the stunting prevalence by rural/urban residence for the most recent DHS survey. In all cases, the stunting prevalence was lower in the urban than in the rural areas. Stunting prevalence in urban settings ranged from 10% in Senegal to 43% in Madagascar. Stunting prevalence in rural settings ranged between 20% in Senegal and 59% in Burundi.

Similarly, the wealthiest quintile had the lowest prevalence of stunting (Figure 5). Countries like Cameroon, Burundi, and Nigeria had a high spread of stunting prevalence between the poorest and the wealthiest quintile. A difference in stunting prevalence of 35–38% was observed between the poorest and the wealthiest quintiles.

### 3.4. Access to Essential Health Care and Child Feeding Practices (Diet Quality)

Compared to the urban and the wealthiest quintile, the poorest quintile and the rural had a lower CCI and proportion of children meeting the MDDS (*p* < 0.01; Figure 6). Only 2 in 5 children met the MMF, about 1 in 5 met the MDDS, and less than 1 in 10 met the criteria for MAD (Appendix A).

## 4. Discussion

The present study highlights that the stunting prevalence has declined in most parts of the continent, but the pace of change was not the same everywhere in Africa. In about half of the studied countries, faster rates of change were observed in the most disadvantaged communities (rural and poorest quintile). Despite these changes, the stunting prevalence remains unacceptably high, with significant disparities by wealth quintile and rural-urban residence. These disparities are also illustrated in children’s diet quality and access to health care.

One in three or 58.8 million African children younger than five years of age are stunted. This means that a substantial proportion of African children will have compromised physical and cognitive development, which further undermines their health, education, productivity, and future earnings [4,8]. This is estimated to cost African countries about ~10% of their GDP per capita [7], but this human capital loss may not be equally bared by all segments of the population. Indeed, our stratification of stunting prevalence by rural-urban and wealth quintile illustrates that the poorest quintiles and rural communities are disproportionately affected. Some signs of narrowing inequalities through faster stunting reductions in the most vulnerable groups (poorest quintile and rural residence) were noticed, but in only about half of the studied countries.

Inequalities in stunting prevalence can have multiple reasons and may be symptoms of inequitable resource-sharing and governance, or can be related to harsh environmental conditions, conflicts, etc. [16,17]. Indeed, in many LMIC, the poorest segment of the population remains marginalized and suffer from poor infrastructure, low income, gender inequality, and limited access to education and health services, which entraps it into a vicious cycle of poverty and poor health [18,19]. The coverage of essential curative and preventive RMNCH interventions (CCI) confirms this inequality in the access to the most basic health care, which can partly be related to insufficient funding to deliver interventions at scale. This is particularly the case for nutrition interventions, as for example in 2015, high stunting burden countries only invested 1% (2.9 billion USD) of their annual health budgets on nutrition-specific programs, illustrating a significant resource gap compared to the additional ~5 billion USD needed annually to achieve the WHA stunting targets [20].

Besides access to health care, a key determinant of stunting is the children’s complementary diet [21]. The complementary feeding period (6–23 months) coincides with the timing of growth faltering, as suboptimal complementary feeding, both in quantity and quality, do not fulfill the high energy and nutrient required to sustain the rapid growth and development expected in this period [21,22]. Although poor complementary diets were observed for children from all segments of the population, the rural and the poorest wealth quintiles were again the most affected. The poor complementary feeding practices can be associated with poor nutritional-literacy, but can also reflect constraints in the availability, accessibility, and affordability of nutrient-dense foods [23,24,25].

Ensuring adequate supply of food, both in quantity and quality, is critical to improving diets. Although agricultural production has substantially increased in many African countries, agricultural policies have focused on cereal yields, leading to supply deficits of nutrient-dense foods, which are partly reflected in the prices of such foods [26]. For example in Ethiopia, prices of nutrient-dense foods (e.g., fruits) increased faster than the national inflation rate in the last decade [27]. Headey and Alderman [28] also show how relative prices of animal source foods, which are critical for preventing stunting [29], are much more expensive in Africa than elsewhere. Addressing socio-economic disparities in complementary feeding would thus require not only increasing coverage of nutrition education, but also calls for food systems innovations that can make nutrient-dense foods accessible and affordable for all. Aligning health and food systems is thus needed to create the much-needed synergy to increase both demand and supply of healthy diets.

## 5. Strength and Limitations

The use of the largest available, nationally representative, and mutually comparable cross-sectional DHS and MICS surveys is a key strength of this study. The disaggregation of stunting, health coverage, and complementary diets by socio-economic indicators such as wealth quintile and rural/urban residence helped unmask the disparities that would otherwise be hidden in aggregate, national-level estimates. However, the following limitations need to be considered when interpreting our findings. Two or more rounds of data were not available for all the countries and thus we were only able to provide trends for a limited number of countries. In addition, the cross-sectional nature of the study does not allow any causal inferences to be made. The CCI is composed of interventions that may not be directly related to nutrition outcomes but were used as a proxy of health care access. Because of the complex and multifactorial causes of stunting, this study is ill-equipped to explain observed disparities between countries. This calls for future studies that systematically investigate observed disparities in selected countries, possibly through regression decomposition approaches.

## 6. Conclusions

Despite decreases in the prevalence of stunting over the past decade, the reductions in stunting were uneven between and within the countries. Stunting disproportionately affected the rural and the poorest wealth quintile. These inequalities are also observed in access to health care and the quality of complementary foods, suggesting that regional and national level estimates, although useful for monitoring and tracking progress, mask disparities by socio-economic status. Therefore, monitoring progress by socioeconomic status is critical to inform the design and implementation of interventions aiming to prevent stunting. Adequate and predictable funding for nutrition interventions, but also aligning food- and health-systems’ interventions are needed to accelerate stunting reduction. Given the long-term adverse effects of stunting, attention to equity in stunting prevention efforts is critical to prevent the widening of inequalities between and within countries.

## Figures and Tables

**Figure 1 nutrients-12-00253-f001:**
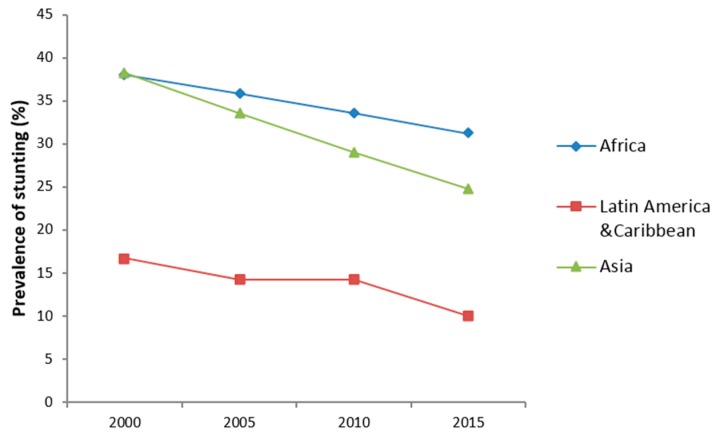
Average stunting reductions in regions with Low- and Middle-Income Countries (LMIC). Authors’ graph using UNICEF-WHO-The World Bank: joint malnutrition estimates [2].

**Figure 2 nutrients-12-00253-f002:**
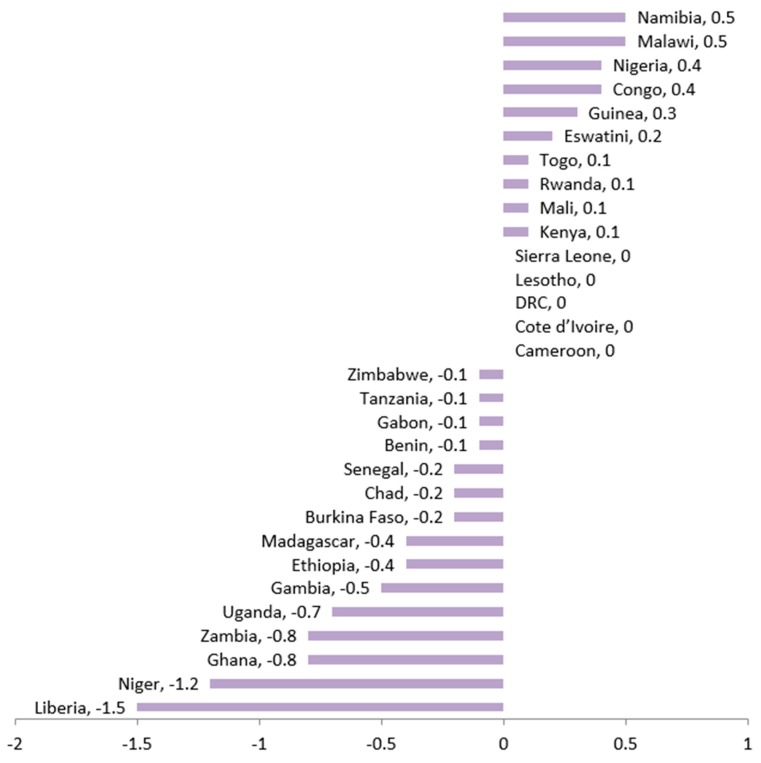
Change of stunting prevalence in rural relative to urban residence (rural-urban) over 1998–2008 to 2009–2018 periods. The graph presents excess changes relative to urban residence; negative values are desirable outcomes as it means faster changes in stunting are happening in the most disadvantaged group.

**Figure 3 nutrients-12-00253-f003:**
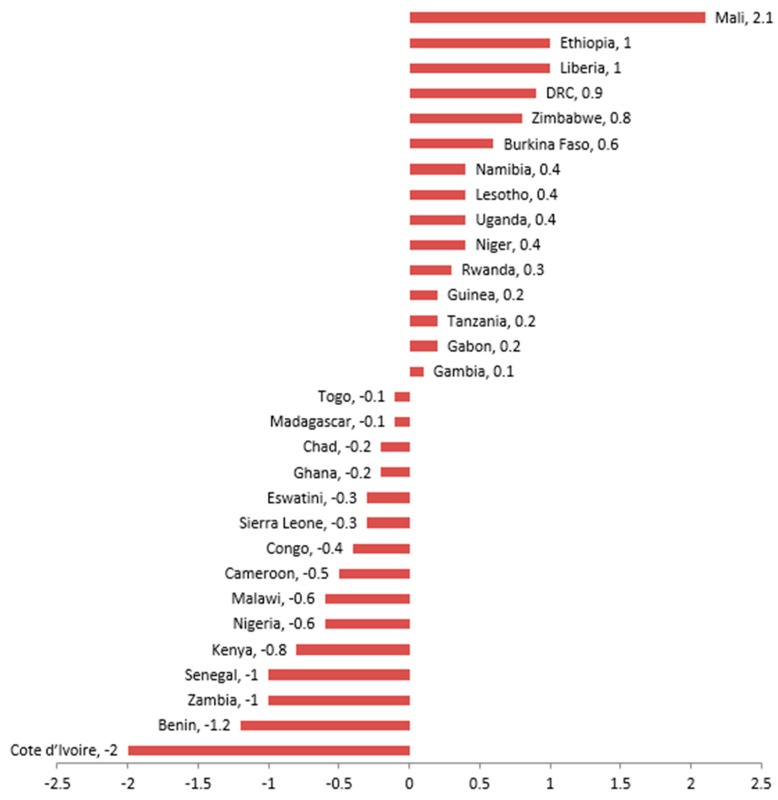
Change of stunting prevalence in the poorest relative to the wealthiest quintile (poorest-richest) over 1998–2008 to 2009–2018 periods. The graph presents excess changes relative to the wealthiest quintile; negative values are desirable outcomes as it means faster changes in stunting has happened in the most disadvantaged group.

**Figure 4 nutrients-12-00253-f004:**
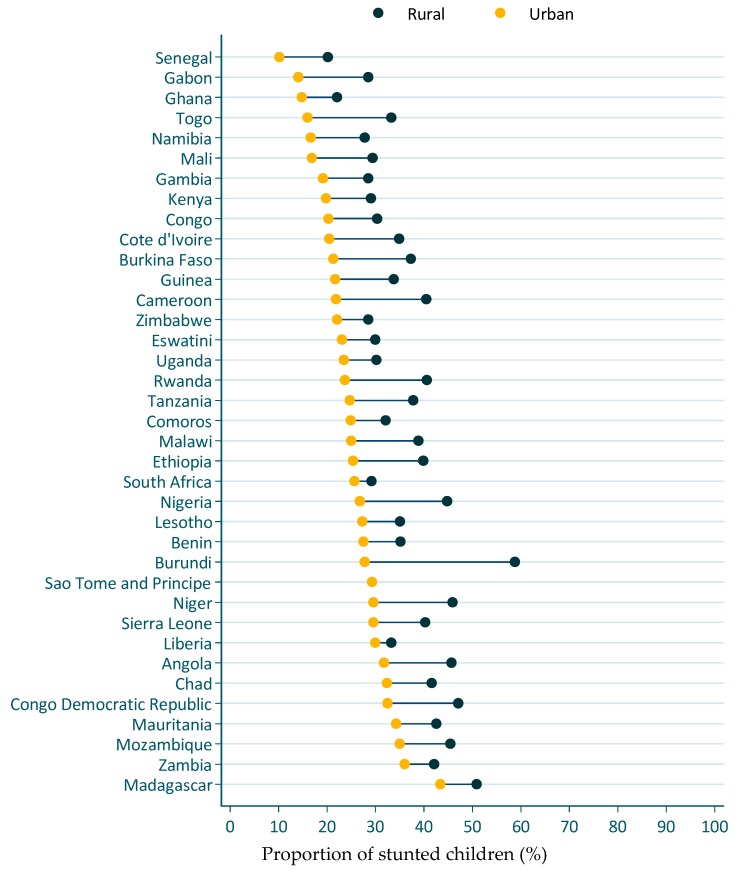
Stunting prevalence by rural/urban residence. Data is from the most recent demographic and health surveys. Dots show stunting prevalence estimates for children younger than five years of age residing in rural (navy blue) and urban (orange) residence.

**Figure 5 nutrients-12-00253-f005:**
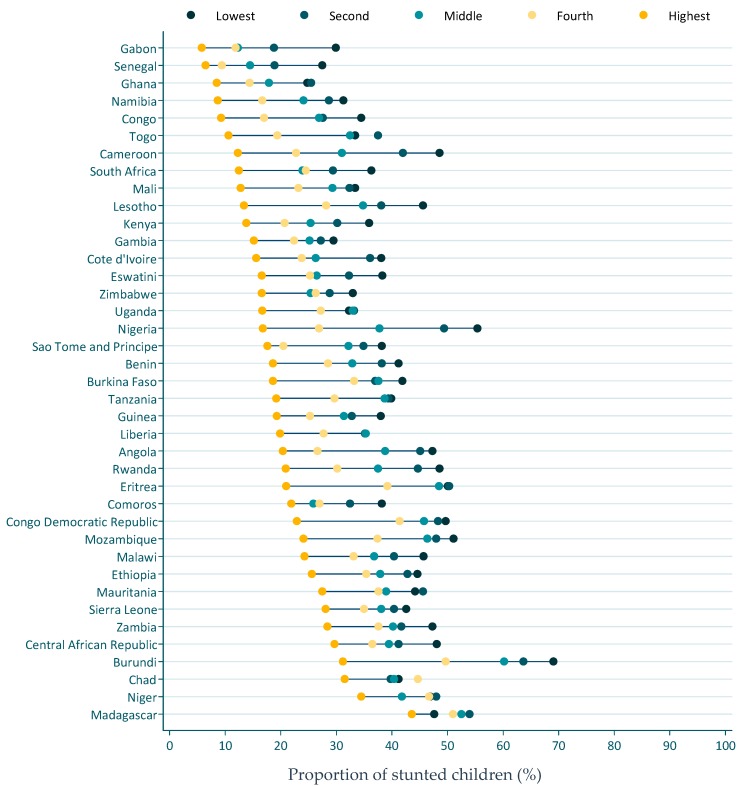
Stunting prevalence by wealth quintile. Dots represent estimated stunting prevalence for the poorest, second, middle, fourth, and the highest wealth quintile.

**Figure 6 nutrients-12-00253-f006:**
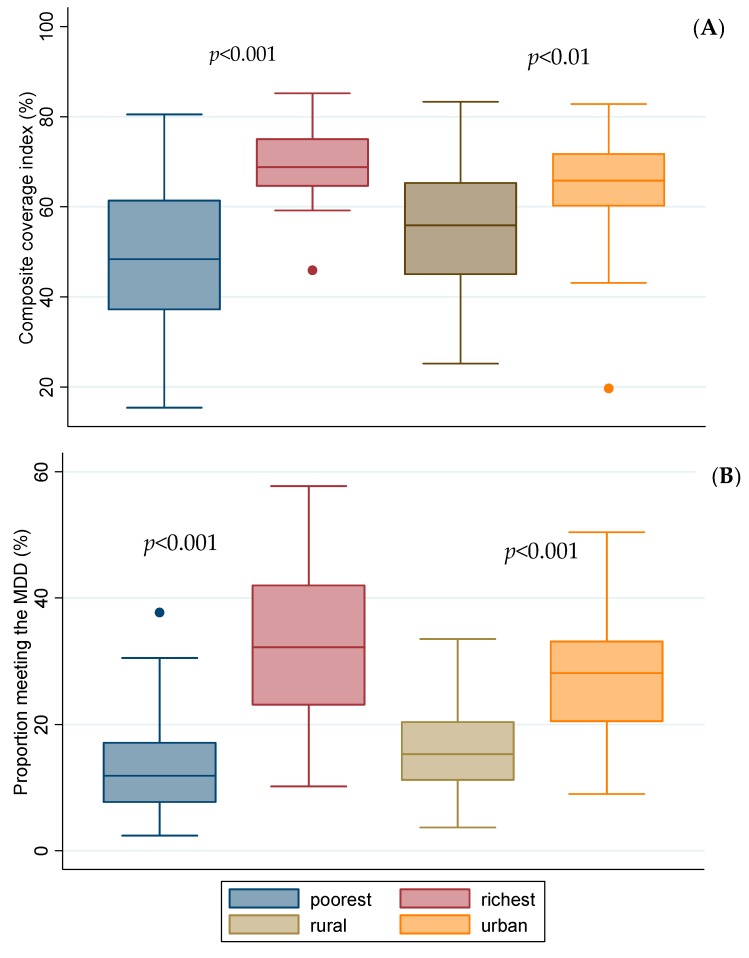
Coverage index of eight reproductive maternal neonatal and child interventions along the continuum of care (**A**) and proportion of children meeting minimum dietary diversity score (**B**) by rural/urban and poorest/richest quintile (*n* = 34). The boxplots present country level analyses of the latest round of DHS or MICS. Statistical tests are from independent *t*-test comparing prevalence in the poorest to the richest quintiles and rural to urban residents.

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
