# Peer review of "Socio-Economic Inequalities in Child Stunting Reduction in Sub-Saharan Africa"

_nutrients, 2020, doi:10.3390/nu12010253_

Round 1

Reviewer 1 Report

Socio-economic inequalities in child stunting reduction in Sub-Saharan Africa

Baye et al. aimed to evaluate the access of essential health interventions, diet quality, and stunting by wealth quintile and rural/urban residence.

The paper is simple but original and brings us information about stunting in Sub-Saharan Africa which is not widely available. The paper uses data from nationally representative Demographic and Health Services and other indicators available in Africa. This gives the paper a high validity.

Overall the presentation is accurate and the conclusions are supported by the data. However, there are several minor points to be edited:

P 2, L47. DEfine DHS

P 2, L56 Change calculated for estimated

P 2, L62 Move CCI to L63

P 3, L100-101 Cancel one "using"

P 3, L104 Is it <0.05 or ≤0.05

Figure 6 Please add A and B to the graphic. They are only present in the caption

The text should be edited by a native English speaker as there are some sentences not clear

Author Response

Thank you for the constructive comments. "Please see the attachment" for a point-by-point response. 

Reviewer 2 Report

This is an important study evaluating current trends of stunting in children in sub-saharan Africa.

The paper can be improved by addressing the following:

Introduction: Add a paragraph on the the prevalence of stunting in other continents and contrast this with the sub-saharan African. Add another paragraph introducing the various known causes of stunting and approaches currently recommended to prevent and control stunting Figure 1: shows data for only two time-points (2000 vs. 2016); this figure must be improved by adding at least two additional time-points (e.g., 2006 and 2010 or 2011) showing how the trends are changing. Figure 2. Data shows at least 10 fold difference between Liberia, Niger vs. Zimbabwe, and 3 other countries; authors must explain possible causes for this huge difference. Same thing for other set of countries: Cameron vs. Namibia: what are the potential reasons? Figure 3: Same issue as in Figure 2 must be explained; why there is more than a 10-fold difference between countries showing the same trends? Figures 4. The figure legend needs more explanation on what point you are making from this dataset? Why the rural plus urban stunting prevalence does not add up to 100%? Figure 5. Same comments as for Figure 4. Figure 6. The legend should contain statements illustrating take home messages from these data sets. Also specify the name of statistical test used for obtaining the p value shown. Results section: Currently, it does not show any structure. Structure the results section with sub-headings; each sub-heading should state the take-home message from each result data sets. For example for each of the Figures, one sub-heading of results section must be used. Discussion: Explain more on how you would 'align food and health -systems interventions'? Is it not being done currently? why/why not? what are the possible barriers? Can authors hypothesize on how many decades it could take to solve this massive stunting problem in the sub-saharan Africa? What type of resources are needed to eliminate this problem? Some funding $ estimates currently invested to address this problem between 2000 vs. 2016 would provide good perspectives on the size of this problem; and also would be helpful to assess the funding needs so that lagging countries could show progress in a decade?

Author Response

Thank you for the constructive comments. Please see the attachment for a point-by-point response.

Round 2

Reviewer 2 Report

The manuscript looks much nicer now!